

# Measurement error associated with gait cycle selection in treadmill running at various speeds

Aaron S. Fox[1], Jason Bonacci[1], John Warmenhoven[2,3] and Meghan F. Keast[1]

[1] Centre for Sport Research, School of Exercise and Nutrition Sciences, Deakin University, Geelong, Victoria, Australia

[2] University of Canberra Research Institute of Sport & Exercise (UCRISE), University of Canberra, Canberra, Australia

[3] Research & Enterprise, University of Canberra, Canberra, Australia

Corresponding author
Aaron S. Fox, aaron.f@deakin.edu.au

## ABSTRACT

A common approach in the biomechanical analysis of running technique is to average data from several gait cycles to compute a 'representative mean.' However, the impact of the quantity and selection of gait cycles on biomechanical measures is not well understood. We examined the effects of gait cycle selection on kinematic data by: (i) comparing representative means calculated from varying numbers of gait cycles to 'global' means from the entire capture period; and (ii) comparing representative means from varying numbers of gait cycles sampled from different parts of the capture period. We used a public dataset ($n = 28$) of lower limb kinematics captured during a 30-second period of treadmill running at three speeds (2.5 m s$^{-1}$, 3.5 m s$^{-1}$ and 4.5 m s$^{-1}$). 'Ground truth' values were determined by averaging data across all collected strides and compared to representative means calculated from random samples (1,000 samples) of $n$ (range = 5–30) consecutive gait cycles. We also compared representative means calculated from n (range = 5–15) consecutive gait cycles randomly sampled (1,000 samples) from within the same data capture period. The mean, variance and range of the absolute error of the representative mean compared to the 'ground truth' mean progressively reduced across all speeds as the number of gait cycles used increased. Similar magnitudes of 'error' were observed between the 2.5 m s$^{-1}$ and 3.5 m s$^{-1}$ speeds at comparable gait cycle numbers —where the maximum errors were < 1.5 degrees even with a small number of gait cycles (*i.e.*, 5–10). At the 4.5 m s$^{-1}$ speed, maximum errors typically exceeded 2–4 degrees when a lower number of gait cycles were used. Subsequently, a higher number of gait cycles (*i.e.*, 25–30) was required to achieve low errors (*i.e.*, 1–2 degrees) at the 4.5 m s$^{-1}$ speed. The mean, variance and range of absolute error of representative means calculated from different parts of the capture period was consistent irrespective of the number of gait cycles used. The error between representative means was low (*i.e.*, < 1.5 degrees) and consistent across the different number of gait cycles at the 2.5 m s$^{-1}$ and 3.5 m s$^{-1}$ speeds, and consistent but larger (*i.e.*, up to 2–4 degrees) at the 4.5 m s$^{-1}$ speed. Our findings suggest that selecting as many gait cycles as possible from a treadmill running bout will minimise potential 'error.' Analysing a small sample (*i.e.*, 5–10 cycles) will typically result in minimal 'error' (*i.e.*, < 2 degrees), particularly at lower speeds (*i.e.*, 2.5 m s$^{-1}$ and 3.5 m s$^{-1}$). Researchers and clinicians should consider the balance between practicalities of collecting and analysing a smaller number of gait cycles against the potential 'error'

when determining their methodological approach. Irrespective of the number of gait cycles used, we recommend that the potential 'error' introduced by the choice of gait cycle number be considered when interpreting the magnitude of effects in treadmill-based running studies.

# INTRODUCTION

Collecting and analysing biomechanical data is frequently used to examine running technique. A common methodological approach is to average data from several gait cycles to compute a given biomechanical measure. Calculating this 'representative mean' is thought to be representative of the individuals' broader running technique. Given the inherent variability in human movement (*Emmerik & Wegen, 2000*), the quantity and selection of gait cycles used to create this 'representative mean' appears an important choice in accurately quantifying an individuals' running gait. However, the number of gait cycles used in biomechanical studies of running varies across the literature (*Oliveira & Pirscoveanu, 2021*). Further, very rarely (if ever) is the decision process underpinning the quantity and selection of gait cycles explained.

It is possible to collect a large number of gait cycles during biomechanical testing, especially during treadmill running. Enabling a participant to settle into a steady gait rhythm may better represent a habitual running pattern. While the collection of a large number of gait cycles can be relatively easy, it is important to give consideration to the analysis of these data. Inflated data cleaning (*e.g.*, labelling and gap filling motion capture data) and analysis (*e.g.*, processing frames *via* inverse kinematics) time occur when processing a running trial that uses many gait cycles. Similarly, trials with many gait cycles require greater data storage access due to larger file sizes. An additional consideration is which gait cycles are selected from within a capture period. Studies often perform an extended capture period where additional gait cycles are collected around those used for analysis (*e.g.*, *Fox, Ferber & Bonacci, 2021*; *Fellin, Manal & Davis, 2010*). The impact of this gait cycle selection on biomechanical outcome measures is yet to be investigated. Better understanding of the impact of gait cycle selection on biomechanical outcome measures may help optimise data collection and analysis practices.

*Oliveira & Pirscoveanu (2021)* examined the typical number of gait cycles used in running biomechanics studies. On average, 12 gait cycles (a gait cycle referring to the period between successive foot-ground contact of a limb) were used to generate biomechanical outcome measures, though very few of these studies (five out of 56) used more than 10 cycles (*Oliveira & Pirscoveanu, 2021*). The impact of sample size (*i.e.,* 10 to 40 runners) and number of gait cycles (*i.e.,* five to 40) used on biomechanical outcome measures (*i.e.,* foot contact time, loading rate, peak vertical ground reaction force, peak braking force, running speed, and foot contact angle) was also examined (*Oliveira & Pirscoveanu, 2021*). The authors found that greater than 10 gait cycles are typically required to achieve stable

biomechanical measures in runners and collecting at least 25 gait cycles will increase the likelihood of achieving stability in the range of biomechanical measures examined (*Oliveira & Pirscoveanu, 2021*). These findings are specific to overground running and the set of biomechanical measures analysed. Treadmill running is often used in research (*Van Hooren et al., 2020*), and it is plausible that the required number of gait cycles required to achieve stability may be different to overground running. Further, Oliveira and Pirscoveanu (*Oliveira & Pirscoveanu, 2021*) did not examine lower limb kinematic variables commonly reported in gait biomechanics studies. These kinematic variables can be presented as both 'zero-dimensional' (0D; *e.g.*, peak values) and 'one-dimensional' (1D; *e.g.*, time-normalised kinematic waveform) variables (*Pataky, Vanrenterghem & Robinson, 2016*). Analyses of these common kinematic variables in both their 0D and 1D forms may provide valuable insight into the number of gait cycles required in biomechanical research. Lastly, Oliveira and Pirscoveanu's (*Oliveira & Pirscoveanu, 2021*) analyses were driven by understanding data stability and statistical significance between two running conditions (*i.e.,* 'normal' *vs.* 'silent' running). A different approach focused on understanding the magnitude of error introduced by analysing different numbers of gait cycles can further our understanding of how gait cycle selection practices impact biomechanical outcome measures. Specifically, understanding the potential error introduced by selecting a different number of gait cycles can aid in interpreting the legitimacy of an effect (*i.e.,* could small effects be due to the set of gait cycles selected).

We sought to extend our current understanding of how the quantity and selection of gait cycles impact lower limb kinematic measures from a 30-s data capture period of treadmill running. First, we examined the magnitude of error introduced in the representative mean compared to the entire bout of treadmill running when the number of gait cycle samples is varied. Second, we examined the potential variation introduced in the representative mean when sampling a set number of gait cycles from different parts of the capture period.

## MATERIALS & METHODS

### Dataset

We used the public dataset of treadmill running biomechanics from *Fukuchi, Fukuchi & Duarte (2017)*. The specifics of this dataset and participant details can be found in the associated paper (*Fukuchi, Fukuchi & Duarte, 2017*). Briefly, this dataset contains lower-extremity kinematics and kinetics of 28 regular (i.e. recreational, injury free) runners (27 male, 1 female; age $= 34.8 \pm 6.7$ years; height $= 1.76 \pm 0.68$ m; mass $= 69.6 \pm 7.7$ kg; running experience $= 8.5 \pm 7.0$ years; running pace $= 4.1 \pm 0.4$ min/km) (*Fukuchi, Fukuchi & Duarte, 2017*). Running kinematics were collected using a 12-camera 3D motion capture system (Raptor-4, Motion Analysis, Santa Rosa, CA, United States) and ground reaction force (GRF) data *via* an instrumented dual-belt treadmill (FIT, Bertec, Columbus, OH, United States) (*Fukuchi, Fukuchi & Duarte, 2017*). Participants ran on the treadmill at three speeds in order ($2.5 \, \text{m s}^{-1}$, $3.5 \, \text{m s}^{-1}$ and $4.5 \, \text{m s}^{-1}$), during which a 3-min accommodation period was provided followed by a 30-s data collection period (*Fukuchi, Fukuchi & Duarte, 2017*).

We processed the raw experimental data from *Fukuchi, Fukuchi & Duarte (2017)* using OpenSim 4.0 (*Delp et al., 2007*). Segment geometry of a generic musculoskeletal model of the pelvis and lower limb provided by *Lai, Arnold & Wakeling (2017)* were scaled for each participant using their static calibration trial, which was also used as a reference for adjusting marker positions on the model. Lower limb joint angles were calculated using filtered (10 Hz low-pass 4th order Butterworth) marker trajectory data within inverse kinematics analysis. GRF data were filtered using the same cut-off frequency and filter. The filtering procedures reflected those originally performed by *Fukuchi, Fukuchi & Duarte (2017)*. Foot strike and toe-off events were determined when the vertical GRF crossed a 20N threshold, also in line with the original work (*Fukuchi, Fukuchi & Duarte, 2017*).

## Data analysis

Kinematic variables common to gait biomechanics studies (*i.e.,* hip flexion/extension, hip adduction/abduction, hip internal/external rotation, knee flexion and ankle plantarflexion/dorsiflexion) were extracted from the right limb for all participants. Data between consecutive foot strikes were extracted and time-normalised to 0–100% of the gait cycle. The time-normalised 1D curves were used in subsequent 1D analyses, while a set of peak variables (hip flexion, hip adduction, hip internal rotation, knee flexion, ankle dorsiflexion) were calculated and extracted for the 0D analyses.

To examine how the number of gait cycles used impacts the representative kinematic mean (*i.e.,* aim 1), we determined 'ground truth' values to compare to for the 0D and 1D kinematic variables by calculating the mean from all available gait cycles in the 30-s capture period of treadmill running. This value was thought to be the most representative of each participant's average running kinematics and was not influenced by the selection of a subset of gait cycles. We then iteratively calculated mean values across the kinematic variables using a range ($n = 5$–$30$) of gait cycles from the data capture period. For each iteration, a random sample of $n$ consecutive gait cycles were extracted and used to calculate a representative kinematic mean. We then compared this representative kinematic mean to the 'ground truth' value for the respective variable to determine the error that gait cycle number selection could introduce.

To examine how sampling gait cycles from different sections of the capture period impacts the representative kinematic mean (*i.e.,* aim 2), we iteratively calculated representative kinematic means using a range ($n = 5$–$15$) of randomly sampled consecutive gait cycles from different parts of the capture period. A smaller range of gait cycles was required for this analysis to avoid sharing gait cycles between the calculated means. For each sampling iteration, we randomly sampled $n$ consecutive gait cycles from two non-overlapping parts of the capture period. We then compared the calculated representative kinematic means between the two parts to determine the error or variation that selection of gait cycles from different parts of the capture period could introduce.

We quantified error in a similar fashion across both aims. For 0D variables, the absolute difference between the representative mean and 'ground truth' (*i.e.,* aim 1) or two representative means (*i.e.,* aim 2) was recorded in each sampling iteration. For 1D variables, the absolute difference between the representative mean and 'ground truth'

(*i.e.,* aim 1) or two representative means (*i.e.,* aim 2) at each normalised sample point across the time-normalised gait cycle were calculated, and the peak difference recorded. The random sampling process for each *n* of gait cycles was repeated 1,000 times for each participant at each running speed—and the error values collated to present descriptive statistics (*i.e.,* mean ± standard deviation (SD), median, range, inter-quartile range) for each gait cycle number across the kinematic variables and running speeds.

## RESULTS

### How does the number of gait cycles used impact the representative kinematic mean?

For the peak 0D kinematic variables, the mean, variance and range of the absolute error of the representative kinematic mean compared to the 'ground truth' mean progressively reduced as the number of gait cycles used increased (see Figs. 1–3). Similar magnitudes of error were observed between the 2.5 m s$^{-1}$ and 3.5 m s$^{-1}$ speeds across the 0D kinematic variables at comparable gait cycle numbers—where the maximum errors were less than 1° even when using a small number of gait cycles (see Fig. S1). The maximum errors at the 4.5 m s$^{-1}$ speed typically exceeded 1−2°, particularly for peak hip and knee joint angles when a lower number of gait cycles were used. Subsequently, a much higher number of gait cycles (*i.e.,* 25–30) were required at 4.5 m s$^{-1}$ to achieve a similar magnitude of error seen at the slower running speeds (see Fig. S1). The larger error values observed at 4.5 m s$^{-1}$ were driven by a bimodal distribution—whereby certain sampling iterations within the same biomechanical measure could produce relatively higher *versus* lower errors (see Fig. 3). The exception to this difference at the higher speed was for peak ankle dorsiflexion, where similarly low error values and ranges (*i.e.,* <0.5°) were observed across all speeds (see Fig. S1).

We observed near identical characteristics of the mean, variance and range of the peak absolute error of the representative kinematic mean compared to the mean from all gait cycles for the 1D kinematic variables (see Figs. 4–6). As with the 0D variables, the potential error reduced as the number of gait cycles increased, and similarly low magnitudes of error (*i.e.,* <1°) were at the 2.5 m s$^{-1}$ and 3.5 m s$^{-1}$ speeds (see Fig. S2). Larger 'errors' exceeding 1−2° with lower gait cycle numbers were present at the 4.5 m s$^{-1}$ speed (with the exception of ankle dorsi/plantarflexion) (see Fig. S2), with this again driven by a more bimodal distribution of samples (see Fig. 6).

### How does the selection of gait cycles impact the representative kinematic mean?

The mean, variance and range of the absolute error (or variation) of the representative kinematic mean compared to the mean from all gait cycles for the peak 0D kinematic variables remained relatively consistent irrespective of the number of gait cycles used when sampling from different parts of the capture period (see Figs. 7–9). At the 2.5 m s$^{-1}$ and 3.5 m s$^{-1}$ speeds, the variation in peak kinematic variables was always less than 1.5° (see Fig. S3). However, peak knee flexion had the potential for larger variation compared to the remaining kinematic variables (see Figs. 7 and 8). While the potential variation between

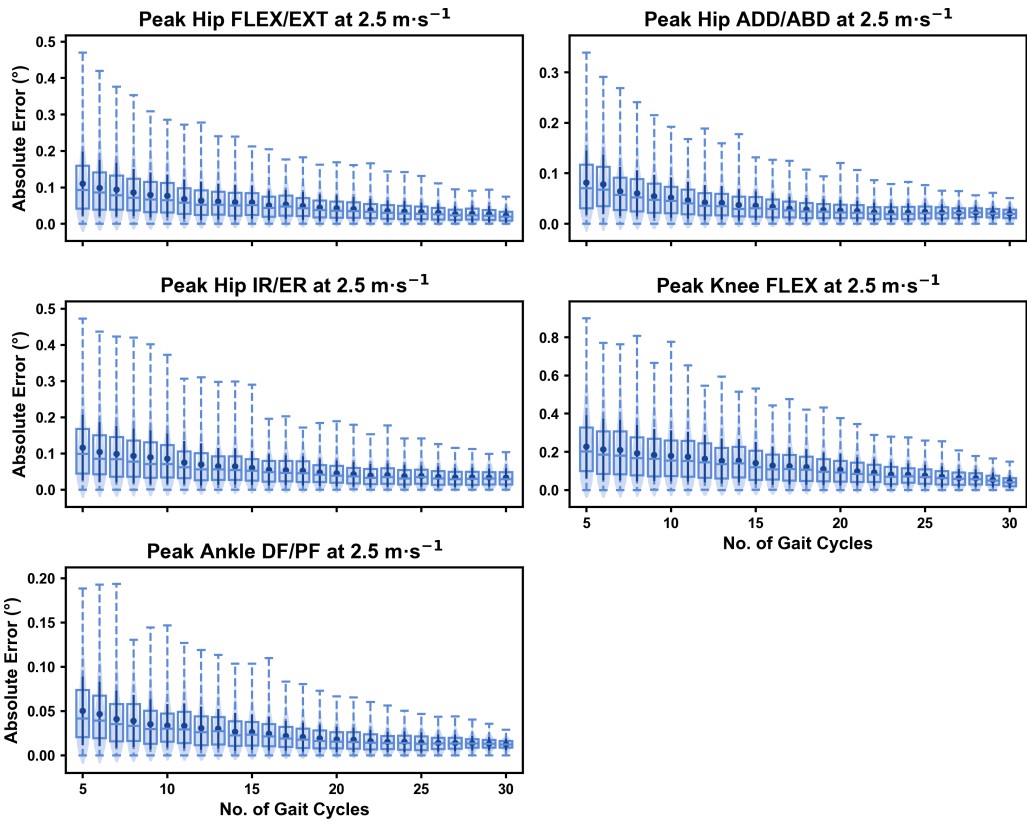

**Figure 1** **Absolute error in peak kinematic variables (*i.e.*, zero-dimensional (0D)) when running at 2.5 m s$^{-1}$ using a subset of gait cycles *versus* all gait cycles from the 30-second treadmill bout.** Darker points and solid lines equate to the mean ± standard deviation. Horizontal lines within boxes equate to the median value, boxes indicate the 25th to 75th percentile, and dashed whiskers indicate the range. Shaded violins are included to illustrate the distribution of values. FLEX, flexion; EXT, extension; ADD, adduction; ABD, abduction; IR, internal rotation; ER, external rotation; DF, dorsiflexion; PF, plantarflexion.

gait cycle samples was consistent with increasing gait cycle numbers at the 4.5 m s$^{-1}$ speed, a higher mean and range of potential variation (*i.e.,* up to 2−4°) was evident across the peak kinematic variables (with the exception of peak ankle dorsiflexion) (see Fig. S3). As in the previous analyses, we observed a bimodal distribution of the samples at the 4.5 m s$^{-1}$ speed (see Fig. 9).

We observed similar characteristics for the mean, variance and range of the absolute error (or variation) of the representative kinematic mean compared to the mean from all gait cycles for the 1D kinematic variables when sampling gait cycles from different parts of the capture period (see Figs. 10–12). The potential variation remained low (*i.e.*, <1.5°) and consistent across the different number of gait cycles at the 2.5 m s$^{-1}$ and 3.5 m s$^{-1}$ speeds (see Figs. 10 and 11, and Fig. S4). The potential variation remained consistent but increased in magnitude (*i.e.*, up to 2−4°), and shifted to a bimodal distribution at the 4.5 m s$^{-1}$ speed (see Fig. 12). In contrast to the 0D variables, this shift was evident in all 1D kinematic variables (including ankle dorsi/plantarflexion).

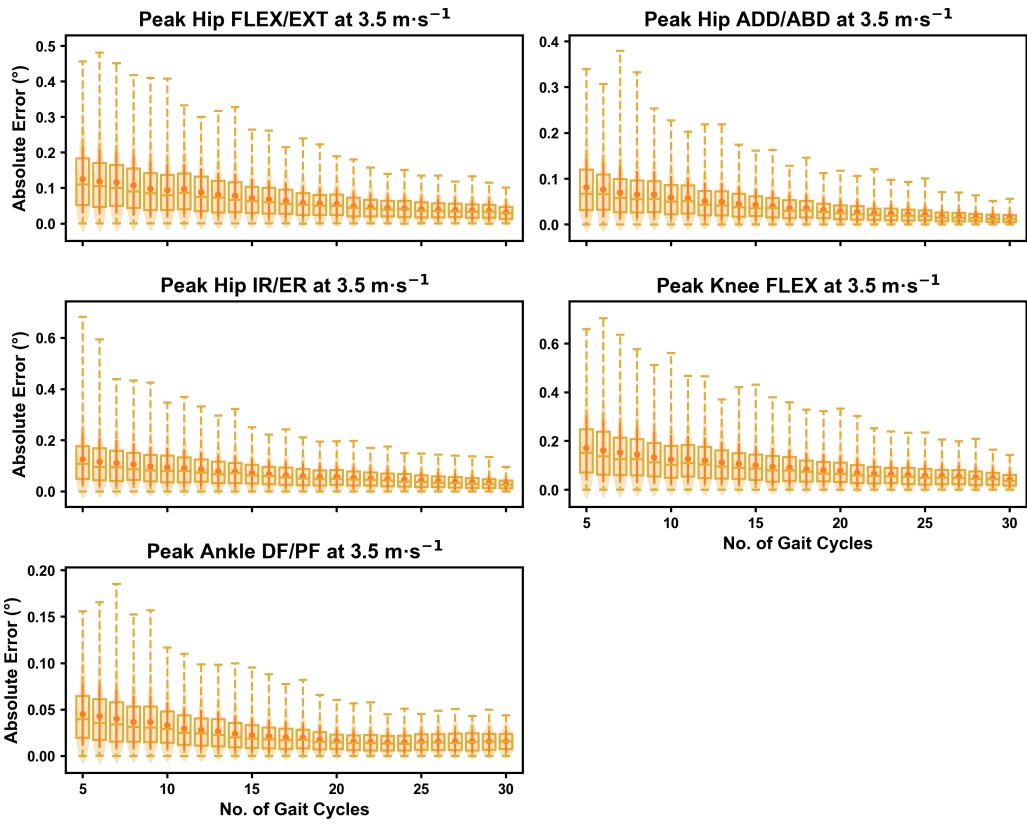

**Figure 2** Absolute error in peak kinematic variables (*i.e.,* zero-dimensional (0D)) when running at 3.5 m s$^{-1}$ using a subset of gait cycles *versus* all gait cycles from the 30-second treadmill bout. Darker points and solid lines equate to the mean ± standard deviation. Horizontal lines within boxes equate to the median value, boxes indicate the 25th to 75th percentile, and dashed whiskers indicate the range. Shaded violins are included to illustrate the distribution of values. FLEX, flexion; EXT, extension; ADD, adduction; ABD, abduction; IR, internal rotation; ER, external rotation; DF, dorsiflexion; PF, plantarflexion.

## DISCUSSION

Biomechanical studies of running often use a subset of gait cycles from a running bout or capture period, and average across these cycles to calculate an individual's representative mean. We examined the impact of the quantity and selection of gait cycles from within a capture period on the magnitude of error in lower limb kinematic measures during a continuous bout of treadmill running. We found that including a greater number of gait cycles to calculate the representative kinematic mean reduces the magnitude and range of potential 'error.' The potential error using a small number of gait cycles (*i.e.,* $n = 5–10$) was low (*i.e.,* typically <1°) when running at 2.5 m s$^{-1}$ and 3.5 m s$^{-1}$, and hence we noted an effect of diminishing returns (*i.e.,* limited improvement in error reduction above 5–10 gait cycles) by including more gait cycles at these slower speeds. Using a similarly low number of gait cycles did slightly inflate potential error (*i.e.,* 1−4°) when running at 4.5 m s$^{-1}$. We also found small magnitudes of error in representative kinematic means across all

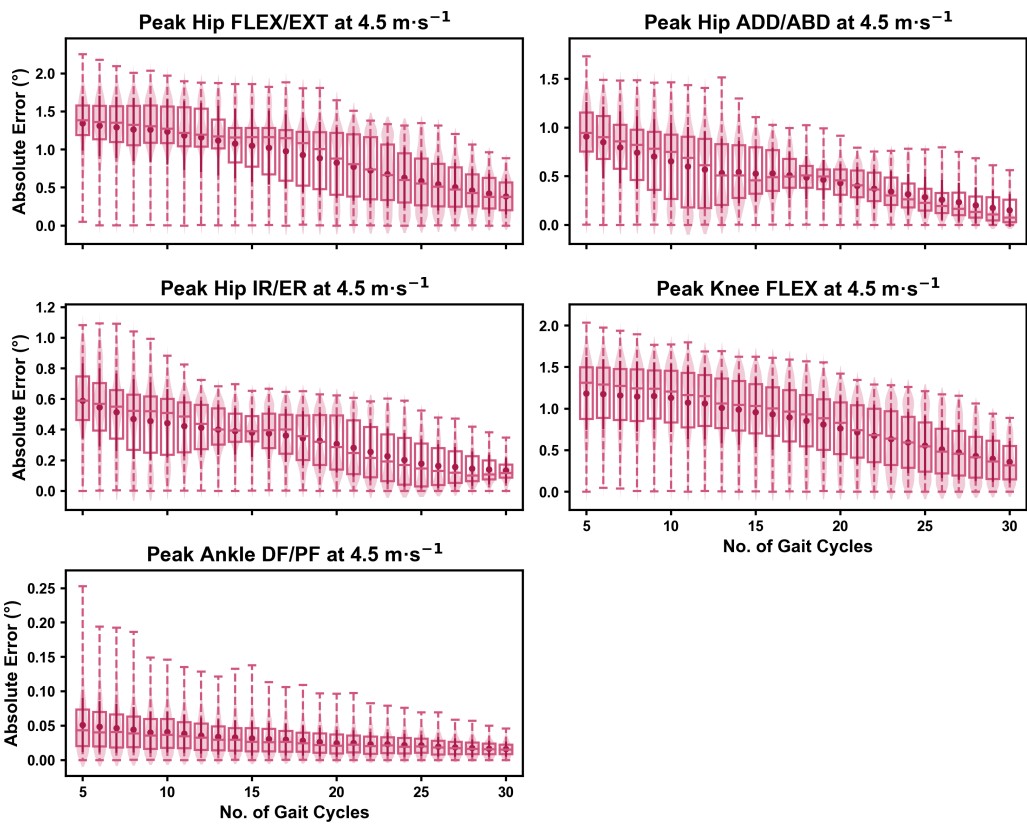

**Figure 3** **Absolute error in peak kinematic variables (*i.e.,* zero-dimensional (0D)) when running at 4.5 m s⁻¹ using a subset of gait cycles *versus* all gait cycles from the 30-second treadmill bout.** Darker points and solid lines equate to the mean ± standard deviation. Horizontal lines within boxes equate to the median value, boxes indicate the 25th to 75th percentile, and dashed whiskers indicate the range. Shaded violins are included to illustrate the distribution of values. FLEX, flexion; EXT, extension; ADD, adduction; ABD, abduction; IR, internal rotation; ER, external rotation; DF, dorsiflexion; PF, plantarflexion.

running speeds when selecting gait cycles from different parts of the capture period, and these remained relatively consistent irrespective of the number of gait cycles used.

We found that the error between the representative kinematic means and the associated 'ground truth' values progressively reduced with an increasing number of gait cycles. Using a greater number of gait cycles equated to using a higher proportion of data that were used to create the 'ground truth'—hence this result is not surprising. More noteworthy is the scale of error when using a reduced number of gait cycles (*i.e., n* = 5–10) and the diminishing effect of using a larger number (*i.e., n* >15) of gait cycles. We typically observed that the maximum error or variation with respect to the 'ground truth' was less than 1°, even at the lowest number of gait cycles used when running at the 2.5 m s⁻¹ and 3.5 m s⁻¹. This error increased up to 3° at 4.5 m s⁻¹. Reducing the potential error compared to the ground truth appeared to be the main effect of increasing the number of gait cycles used. However, the reduction in potential error typically plateaued and a diminished benefit observed when using above 15–20 gait cycles. These patterns were consistent across both

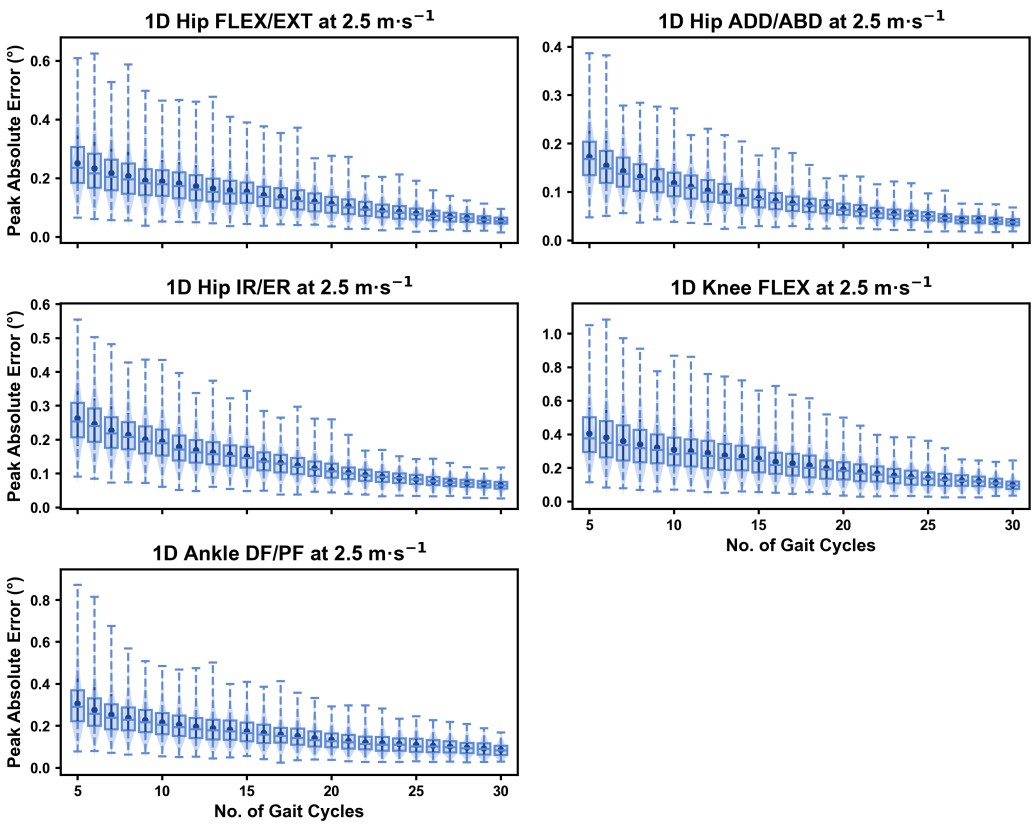

**Figure 4** Peak absolute error in kinematic variables across the gait cycle (*i.e.,* one-dimensional (1D)) when running at 2.5 m s$^{-1}$ using a subset of gait cycles *versus* all gait cycles from the 30-second tread-mill bout. Darker points and solid lines equate to the mean ± standard deviation. Horizontal lines within boxes equate to the median value, boxes indicate the 25th to 75th percentile, and dashed whiskers indicate the range. Shaded violins are included to illustrate the distribution of values. FLEX, flexion; EXT, extension; ADD, adduction; ABD, abduction; IR, internal rotation; ER, external rotation; DF, dorsiflexion; PF, plantarflexion.

the 0D and 1D kinematic data. The notion of diminishing returns above 15–20 gait cycles contrasts with the findings of *Oliveira & Pirscoveanu (2021)*—whereby data stability was not achieved in most runners using this number of gait cycles. Clear differences between our study and this existing work (*Oliveira & Pirscoveanu, 2021*) were the metrics used to define error or stability, the biomechanical measures analysed (*i.e.,* joint kinematics *vs.* mostly kinetic variables), and the use of treadmill (including a 3-minute familiarisation period) *versus* overground running. The latter may represent an important distinction, whereby the familiarisation period combined with the more continuous approach of treadmill running led to participants settling into a more stable rhythm during the data capture period. *Forrester (2015)* performed a series of simulations using a similar sequential analysis technique to *Oliveira & Pirscoveanu (2021)* to determine the number of trials required for biomechanical measures with generic means and standard deviations. This work (*Forrester, 2015*) proposed that nine (±8) trials were required to achieve stability of the mean, which is more in line with our findings. We saw the largest potential 'errors' in hip and knee flexion when using a smaller number of gait cycles and this is not surprising given these measures

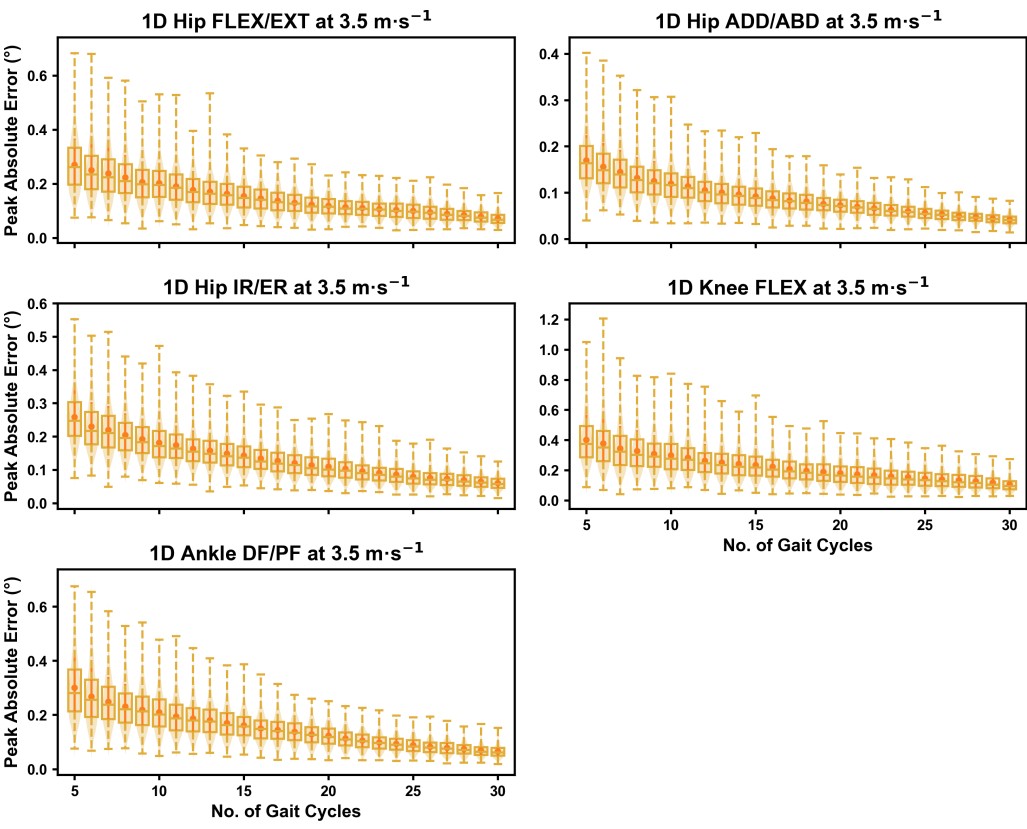

**Figure 5 Peak absolute error in kinematic variables across the gait cycle (*i.e.,* one-dimensional (1D)) when running at 3.5 m s$^{-1}$ using a subset of gait cycles *versus* all gait cycles from the 30-second treadmill bout.** Darker points and solid lines equate to the mean ± standard deviation. Horizontal lines within boxes equate to the median value, boxes indicate the 25th to 75th percentile, and dashed whiskers indicate the range. Shaded violins are included to illustrate the distribution of values. FLEX, flexion; EXT, extension; ADD, adduction; ABD, abduction; IR, internal rotation; ER, external rotation; DF, dorsiflexion; PF, plantarflexion.

had the largest means and standard deviations within the dataset (*Fukuchi, Fukuchi & Duarte, 2017*).

Despite the potential for diminishing returns, our data suggests that researchers can minimise the potential error in representative kinematic means by using more gait cycles. A simplistic recommendation from our analyses would be to use as many gait cycles as possible. However, this ignores the practical considerations of storing, cleaning and processing larger biomechanical data files. Our recommendation is to balance the practical considerations against the potential error or variation in the data that can be tolerated. Consideration should be given to the accuracy of the measure, or size of the effect researchers or clinicians are interested in measuring. For example, using less than ten gait cycles to explore a small effect (*i.e.,* <1−2°) in 1D hip or knee flexion continua would be unwise, as the potential variation in the calculated means could exceed the magnitude of the effect of interest.

We observed relatively small variations (*i.e.,* <1.5°) between representative kinematic means calculated from gait cycle samples extracted from different parts of the 2.5 m

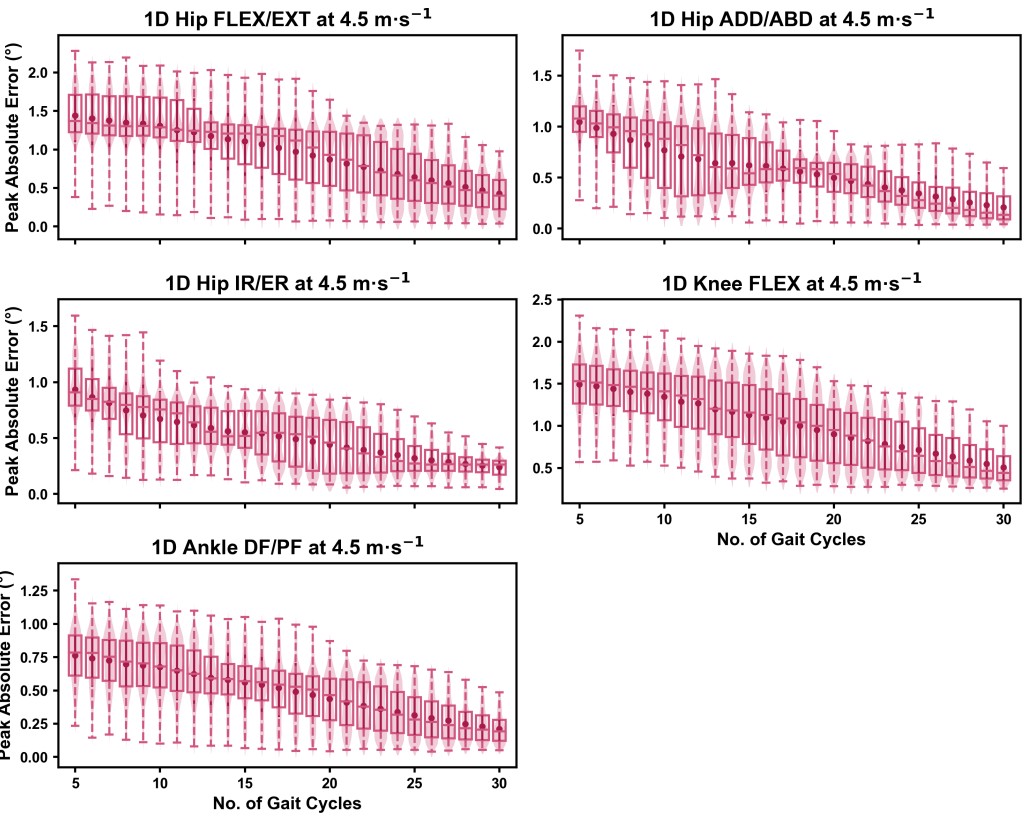

**Figure 6 Peak absolute error in kinematic variables across the gait cycle (*i.e.,* one-dimensional (1D)) when running at 4.5 m s⁻¹ using a subset of gait cycles *versus* all gait cycles from the 30-second treadmill bout.** Darker points and solid lines equate to the mean ± standard deviation. Horizontal lines within boxes equate to the median value, boxes indicate the 25th to 75th percentile, and dashed whiskers indicate the range. Shaded violins are included to illustrate the distribution of values. FLEX, flexion; EXT, extension; ADD, adduction; ABD, abduction; IR, internal rotation; ER, external rotation; DF, dorsiflexion; PF, plantarflexion.

s⁻¹ and 3.5 m s⁻¹ capture periods, while these slightly increased (*i.e.,* 2−4°) when examining the 4.5 m s⁻¹ speed. This magnitude of variation remained consistent irrespective of the total number of gait cycles used. These findings suggest that once the number of gait cycles used for analysis is selected, the selection of these from within a capture period will introduce a small, but consistent amount of 'error.' It is plausible that greater variation could be seen during longer capture periods than that used in the present study (*i.e.,* 30-s) or when comparing gait cycles from capture periods separated by a longer time period (*e.g.,* two capture periods at either end of a 5+ min running bout). The dataset we used did not allow for these analyses to be conducted, yet present relevant avenues for further research on this topic.

Running at 4.5 m s⁻¹ induced greater error relative to the 'ground truth' and between representative means from different parts of the capture period compared to running at 2.5 m s⁻¹ and 3.5 m s⁻¹. There are various potential reasons for these results. Faster running speeds induce larger means and standard deviations across kinematic variables (*Fukuchi, Fukuchi & Duarte, 2017*), particularly in hip and knee flexion where more dramatic

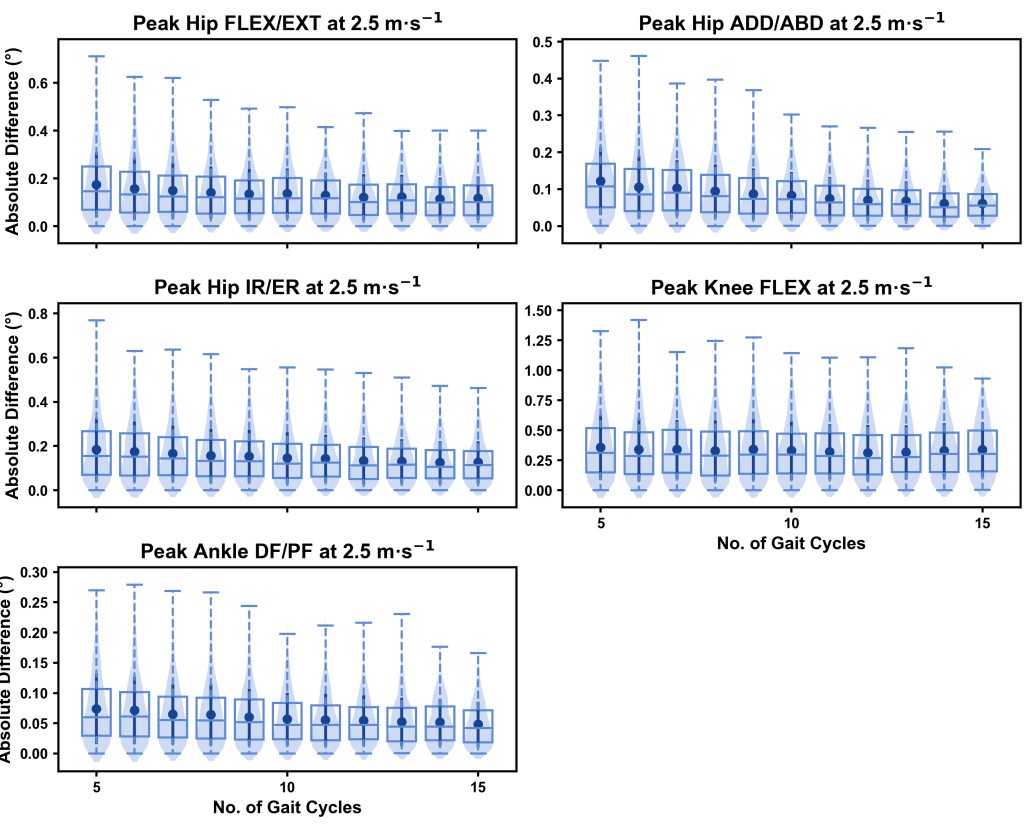

**Figure 7** Absolute error in peak kinematic variables (*i.e.*, zero-dimensional (0D)) when running at 2.5 m s$^{-1}$ using a two comparative subsets of gait cycles from the 30-second treadmill bout. Darker points and solid lines equate to the mean $\pm$ standard deviation. Horizontal lines within boxes equate to the median value, boxes indicate the 25th to 75th percentile, and dashed whiskers indicate the range. Shaded violins are included to illustrate the distribution of values. FLEX, flexion; EXT, extension; ADD, adduction; ABD, abduction; IR, internal rotation; ER, external rotation; DF, dorsiflexion; PF, plantarflexion.

increases in error were observed. An increase in gait speed could also be considered a changed task constraint on the running movement (*Newell, 1985*), and this change in constraint could have affected the role and magnitude of variability. Within the dataset examined, participants ran for a three minute accommodation period at each speed, following which data were collected over a 30-s period (*Fukuchi, Fukuchi & Duarte, 2017*). The order of running conditions (*i.e.*, 2.5 m s$^{-1}$, 3.5 m s$^{-1}$, 4.5 m s$^{-1}$) was kept consistent for each participant (*Fukuchi, Fukuchi & Duarte, 2017*). It is possible that running at the fastest speed towards the end of the running period could have introduced some fatigue when running at 4.5 m s$^{-1}$. If fatigue was present during the final bout of running, it could have resulted in greater kinematic variability (*Chen et al., 2022*; *Meardon, Hamill & Derrick, 2011*) or a change in running kinematics (*Mizrahi et al., 2000*; *Bazuelo-Ruiz et al., 2018*; *Derrick, Dereu & McLean, 2002*) during the 30-s period of data collection.

It is important to note that the measurement error or variation based on gait cycle selection were small compared to other established sources of error during biomechanical data collection and analysis (*Ceseracciu, Sawacha & Cobelli, 2014*; *Leardini et al., 2005*;

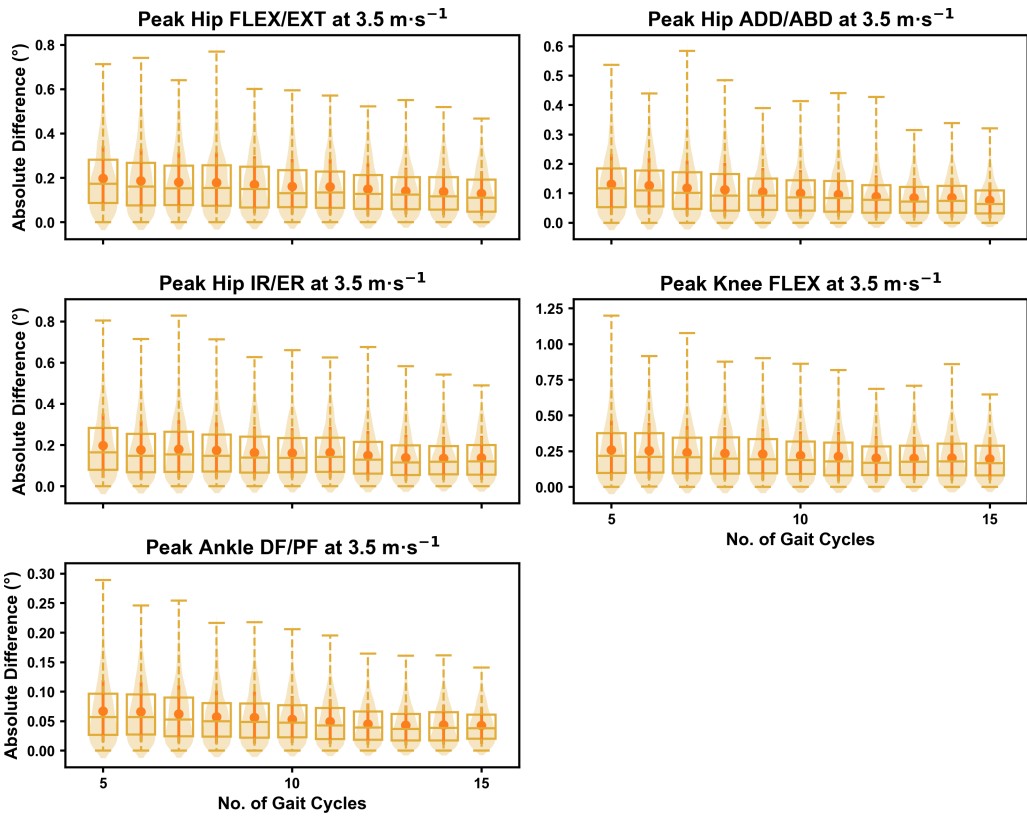

**Figure 8** Absolute error in peak kinematic variables (*i.e.,* zero-dimensional (0D)) when running at 3.5 m s$^{-1}$ using a two comparative subsets of gait cycles from the 30-second treadmill bout. Darker points and solid lines equate to the mean ± standard deviation. Horizontal lines within boxes equate to the median value, boxes indicate the 25th to 75th percentile, and dashed whiskers indicate the range. Shaded violins are included to illustrate the distribution of values. FLEX, flexion; EXT, extension; ADD, adduction; ABD, abduction; IR, internal rotation; ER, external rotation; DF, dorsiflexion; PF, plantarflexion.

*Benoit, Damsgaard & Andersen, 2015*; *Fiorentino et al., 2017*; *Baudet et al., 2014*; *Colle et al., 2016*; *Sauret et al., 2016*; *Mentiplay & Clark, 2018*; *Kainz et al., 2016*; *Leigh, Pohl & Ferber, 2014*; *Sinclair, Hebron & Taylor, 2014*). The magnitude of error in the present study is eclipsed by the errors or variation introduced by soft-tissue artefact associated with skin-mounted markers (*D'Isidoro, Brockmann & Ferguson, 2020*; *Leardini et al., 2005*; *Benoit, Damsgaard & Andersen, 2015*; *Fiorentino et al., 2017*), different joint coordinate systems (*Sauret et al., 2016*; *Baudet et al., 2014*; *Colle et al., 2016*) or gait models (*Mentiplay & Clark, 2018*), kinematic algorithm choice (*Kainz et al., 2016*), tester experience (*Sinclair, Hebron & Taylor, 2014*; *Leigh, Pohl & Ferber, 2014*), or different measurement approaches (*i.e.,* marker *vs.* marker-less) (*Ceseracciu, Sawacha & Cobelli, 2014*). The number of gait cycles used for analysis is likely less important when considering the size of a measured effect against the potential error or variation introduced by other methodological decisions. Future research should also better define practically meaningful effects for biomechanical outcome measures. Using similar methods to those in other fields for defining the smallest
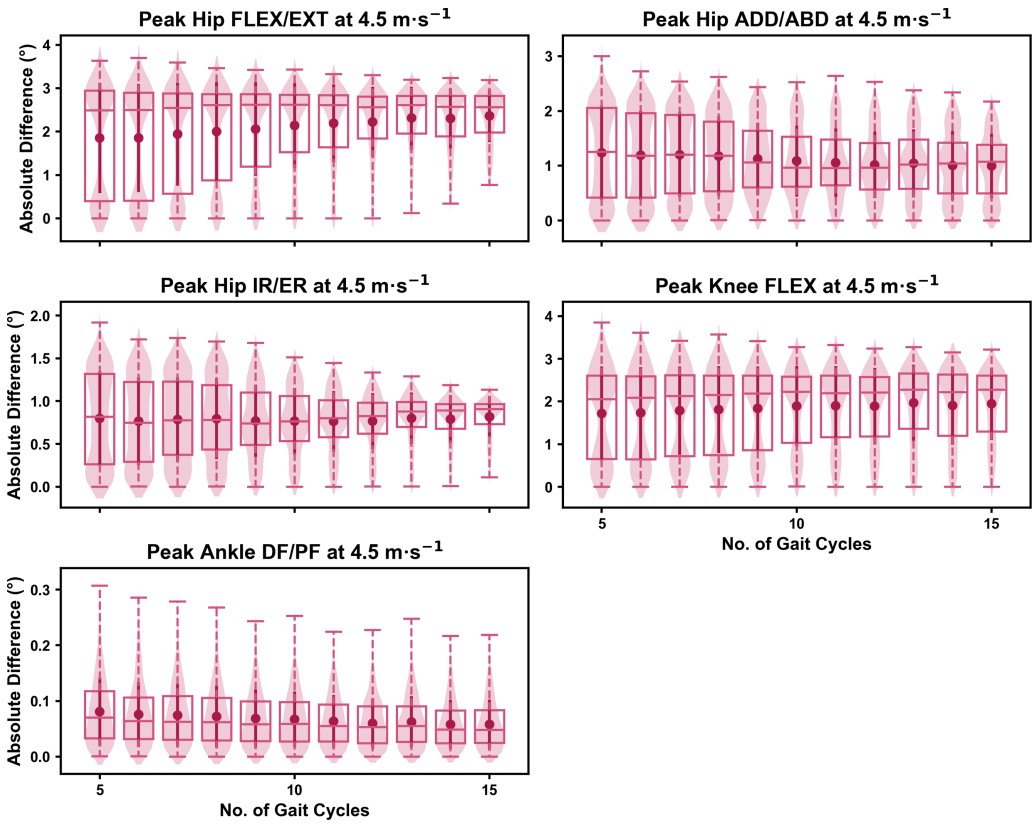

**Figure 9  Absolute error in peak kinematic variables (*i.e.,* zero-dimensional (0D)) when running at 4.5 m s⁻¹ using a two comparative subsets of gait cycles from the 30-second treadmill bout.** Darker points and solid lines equate to the mean ± standard deviation. Horizontal lines within boxes equate to the median value, boxes indicate the 25th to 75th percentile, and dashed whiskers indicate the range. Shaded violins are included to illustrate the distribution of values. FLEX, flexion; EXT, extension; ADD, adduction; ABD, abduction; IR, internal rotation; ER, external rotation; DF, dorsiflexion; PF, plantarflexion.

effect size of interest (*Lakens, Scheel & Isager, 2018*) can help inform whether the magnitude of errors are acceptable for practical use and interpretation.

Our results must be considered with respect to the limitations in our approach. We only examined conditions where *n* consecutive gait cycles were sampled from a 30-s capture period during a continuous bout of treadmill running at three set speeds. Different results might be expected with non-consecutive selection of samples from the capture period, or under different running conditions (*e.g.,* outdoor overground running; slower or faster speeds). We also focused on peak and 1D waveform data of lower limb kinematic variables. Other biomechanical outcome measures (*e.g.,* joint moments, estimates of muscle activation and forces) may incur variable magnitudes of error or variation with respect to gait cycle selection. We inferred error *via* comparison to values calculated from all gait cycles in the running bout (*i.e.,* our 'ground truth' value). Although we deemed this the best approach within our study, it is important to acknowledge that these values may still not represent the individuals' exact or true running kinematics. Lastly, we investigated

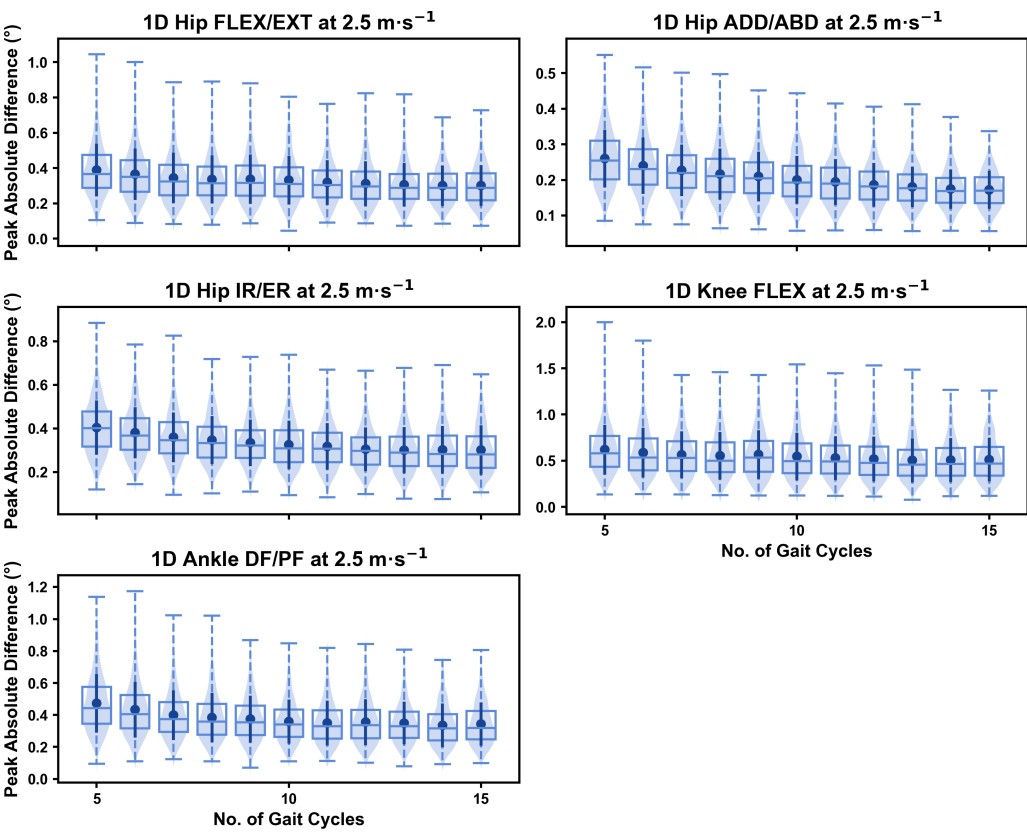

**Figure 10 Peak absolute error in kinematic variables across the gait cycle (*i.e.,* one-dimensional (1D)) when running at 2.5 m s⁻¹ using two comparative subsets of gait cycles from the 30-second treadmill bout.** Darker points and solid lines equate to the mean ± standard deviation. Horizontal lines within boxes equate to the median value, boxes indicate the 25th to 75th percentile, and dashed whiskers indicate the range. Shaded violins are included to illustrate the distribution of values. FLEX, flexion; EXT, extension; ADD, adduction; ABD, abduction; IR, internal rotation; ER, external rotation; DF, dorsiflexion; PF, plantarflexion.

kinematic measures at a univariate joint level. Our findings are therefore not applicable to studies examining covariance or dynamics across joints during gait.

## CONCLUSIONS

We identified the range of potential error or variation in lower limb kinematics associated with the quantity and selection of gait cycles used from a data capture period of continuous treadmill running. Our findings suggest that including as many gait cycles as possible from the running bout will minimise 'error.' However, the error associated with only a small sample of gait cycles (*i.e.,* 5–10 gait cycles) was typically quite small (<3°) when running at 2.5 m s⁻¹ and 3.5 m s⁻¹. Larger potential 'errors' or variation were observed when analysing kinematic variables with larger means and standard deviations, and when running at faster speeds (*i.e.,* 4.5 m s⁻¹). Researchers and clinicians should balance the benefits of a reduction in potential error with the challenges of collecting, processing and analysing a large number

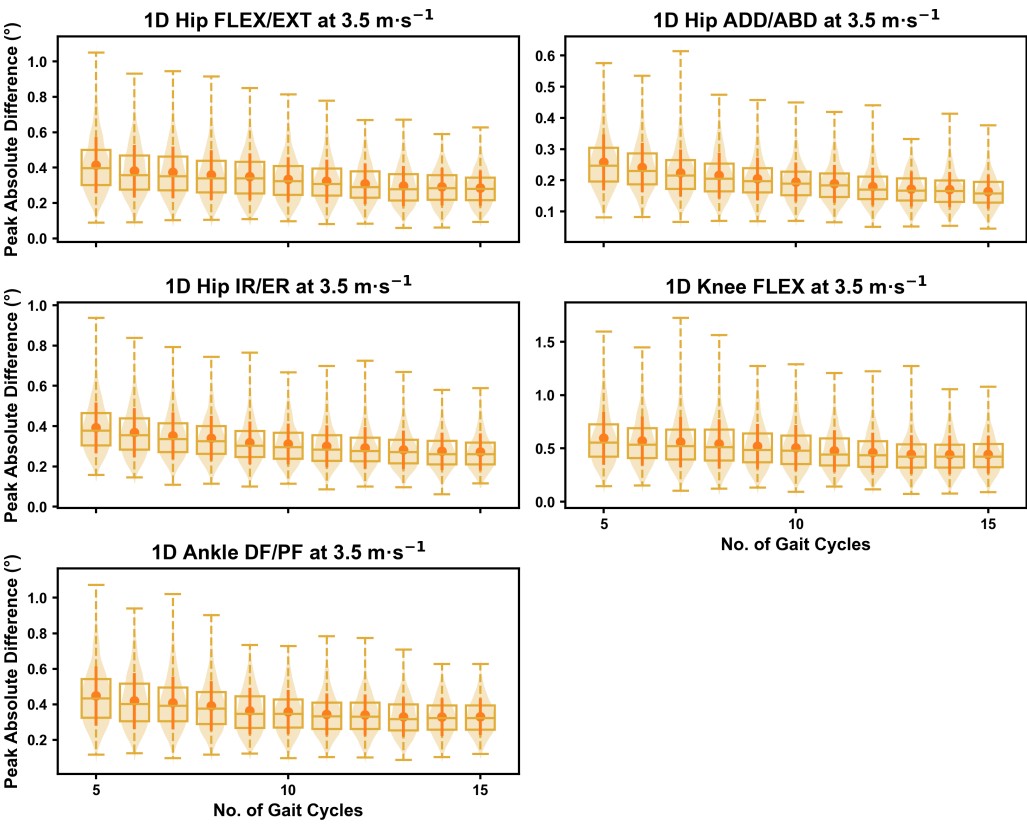

**Figure 11** **Peak absolute error in kinematic variables across the gait cycle (*i.e.,* one-dimensional (1D)) when running at 3.5 m s⁻¹ using two comparative subsets of gait cycles from the 30-second treadmill bout.** Darker points and solid lines equate to the mean ± standard deviation. Horizontal lines within boxes equate to the median value, boxes indicate the 25th to 75th percentile, and dashed whiskers indicate the range. Shaded violins are included to illustrate the distribution of values. FLEX, flexion; EXT, extension; ADD, adduction; ABD, abduction; IR, internal rotation; ER, external rotation; DF, dorsiflexion; PF, plantarflexion.

of gait cycles when determining their methodological approach. We recommend that the potential error or variation introduced by the quantity and selection of gait cycles be considered when interpreting effects from treadmill-based running studies. Specifically, researchers must consider the magnitude of potential error against the identified effects between groups or following an intervention.

### Funding
The authors received no funding for this work.

### Competing Interests
The authors declare there are no competing interests.
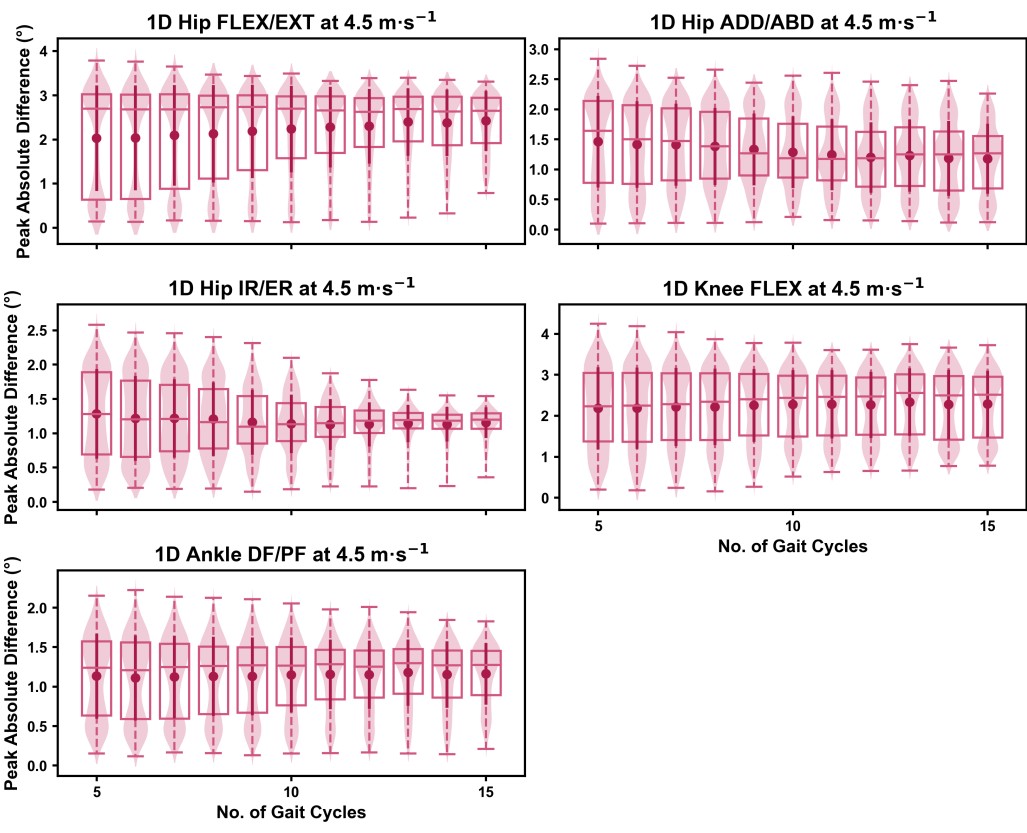

**Figure 12** **Peak absolute error in kinematic variables across the gait cycle (*i.e.,* one-dimensional (1D)) when running at 4.5 m s⁻¹ using two comparative subsets of gait cycles from the 30-second treadmill bout.** Darker points and solid lines equate to the mean ± standard deviation. Horizontal lines within boxes equate to the median value, boxes indicate the 25th to 75th percentile, and dashed whiskers indicate the range. Shaded violins are included to illustrate the distribution of values. FLEX, flexion; EXT, extension; ADD, adduction; ABD, abduction; IR, internal rotation; ER, external rotation; DF, dorsiflexion; PF, plantarflexion.

## Author Contributions

- Aaron S. Fox conceived and designed the experiments, performed the experiments, analyzed the data, prepared figures and/or tables, authored or reviewed drafts of the article, and approved the final draft.
- Jason Bonacci conceived and designed the experiments, authored or reviewed drafts of the article, and approved the final draft.
- John Warmenhoven conceived and designed the experiments, authored or reviewed drafts of the article, and approved the final draft.
- Meghan F. Keast conceived and designed the experiments, authored or reviewed drafts of the article, and approved the final draft.

## Data Availability

The data and analysis code is available at Zenodo: Aaron Fox. (2022). aaronsfox/biomech-trial-selection: Peer J Paper Release (Publication). Zenodo. https://doi.org/10.5281/zenodo.7232338.

## Supplemental Information

Supplemental information for this article can be found online at http://dx.doi.org/10.7717/peerj.14921#supplemental-information.

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
