# Peer review of "Measurement error associated with gait cycle selection in treadmill running at various speeds"

_PeerJ, doi:10.7717/peerj.14921_

## Round 0.1 · original submission · Minor Revisions

· Academic Editor

Minor Revisions

Please, address point-by-point all of Reviewer 2 and 3's raised issues.

·

Basic reporting

I want to congratulate the Authors for this paper. You answered to a typical research question: how may gait cycles I have to analyze? You underlined in a very easy way the answer and the paper is very applicable.

Experimental design

The experimental design is correct and the paper is relevant.

Validity of the findings

The results are interesting and well presented.

Additional comments

No comment

Reviewer 2 ·

Basic reporting

1. Clear, unambiguous and professional usage of English throughout the text.
2. The introduction and the body of the text contain thorough references to previous research, which makes it easy to understand connection to related work.
3. Public datasets are used, allowing the authors’ conclusions to be easily reproducible.
4. Several plots are presented to help understand and explain the statistical basis of the authors’ conclusions.
5. Overall, a very well-written, relatively self-contained and cross-referenced text with clear logical flow.

Experimental design

1. Clear explanation and impact of research outlined. The objective is to verify their hypothesis that including higher number of gait cycles from a running bout will minimize error in lower limb kinematics.
2. The authors provide caveats to the above conclusion that errors are small nevertheless, even for small sample of gait cycles, for lower speeds < 3.5 m/s. However, they show – supported with figures – that their hypothesis more strongly holds for faster speeds > 4.5 m/s. The authors also, very helpfully, make comparison with existing research.
3. They also conclude that from a clinician’s perspective, this is useful information in balancing the diminishing returns between collecting gait cycles and inducing errors.
4. Overall, the experimental design is well-defined from a statistical perspective.

Validity of the findings

1. The raw (open source) data and plots from the authors’ tests are provided, which agree with each other.
2. It is understandable that broad brushstrokes are difficult in this research area. Accordingly, several limitations are outlined, including potential bias due to data from consecutive running bouts, choice of ground truth value, etc.
3. Modulo the above limitations, the conclusions of this paper are solid and relevant from a theoretical and a clinical perspective.

Additional comments

1. While the analysis is comprehensively presented in the figures, it is slightly difficult to compare between kinematic variables at slow and fast speeds. As this is one of the important take-aways of this paper, I would recommend showing a side-by-side comparison.
2. I would suggest that the authors investigate statistical techniques/tests to compare two distributions. I believe that adding results of a T-/Z- test between low-speed and high-speed data would improve readability and credibility.

Reviewer 3 ·

Basic reporting

The discussion is far too long for a study of this nature. I would recommend you shorten it so that the main points are made very clear – as it stands, it is difficult to find the important parts in amongst a lot of unnecessary material. There are some aspects where you could improve the presentation of your writing.

Experimental design

This is quite straightforward, but did you check for outliers in the original sample? This would be worth doing just to make sure that any particular set of gait cycles you analysed did not feature a large outlier that affected that particular sample. This is not unusual in analysis of variability in gait.

Validity of the findings

Line 388 – you mention this before, but have you calculated the errors relative to the magnitude of the variable measures (e.g., the error in degrees relative to the expected range of motion)?

Additional comments

Line 27 (and elsewhere) – please remember to place a space between the numeral and the unit, except for the percentage sign and degree symbol (which confusingly you write as a word in the text but use the symbol in the figures). Also, please use the unit for “second” (s).
Line 58 – please add an apostrophe to “individuals”. Same comment for Lines 60 and 376.
Line 68 – please change “this data” to “these data”.
Lines 79-87 – it would be helpful here to stick to one description of the gait cycle. For example, you mention “gait cycles”, “strides” and “steps” all within this section. It is difficult to follow at times what exactly you mean (you might even want to define a single gait cycle to make this clear).
Line 90 – please change “may be different” to “are different”. Throughout the paper, you use the word “may” a great deal, suggesting that you are not very sure of what you write – check this and rewrite if necessary.
Line 118 – please use the SI unit for height, which is m (rather than cm).
Lines 123 – please change “three-minute” to “3-min” and “30-second” to “30-s”.
Lines 146-155 (and elsewhere) – I understand that we can never fully know the ‘truth’ of a value, but it seems odd that you have ‘ground truth’, ‘most representative’ and ‘error’ in inverted commas. Please remove these.
Line 171 – what is your definition here of a “point”.
Line 348 – please change “that” to “than”.

---

## Round 0.2 · Minor Revisions

· Academic Editor

Minor Revisions

Please, do you your best to address point-by-point all of Reviewer 2's raised issues. Otherwise, I cannot recommend acceptance.

Reviewer 2 ·

Basic reporting

No changes from previous feedback (reproduced below)

1. Clear, unambiguous and professional usage of English throughout the text.
2. The introduction and the body of the text contain thorough references to previous research, which makes it easy to understand connection to related work.
3. Public datasets are used, allowing the authors’ conclusions to be easily reproducible.
4. Several plots are presented to help understand and explain the statistical basis of the authors’ conclusions.
5. Overall, a very well-written, relatively self-contained and cross-referenced text with clear logical flow.

Experimental design

No changes from previous feedback (reproduced below)

1. Clear explanation and impact of research outlined. The objective is to verify their hypothesis that including higher number of gait cycles from a running bout will minimize error in lower limb kinematics.
2. The authors provide caveats to the above conclusion that errors are small nevertheless, even for small sample of gait cycles, for lower speeds < 3.5 m/s. However, they show – supported with figures – that their hypothesis more strongly holds for faster speeds > 4.5 m/s. The authors also, very helpfully, make comparison with existing research.
3. They also conclude that from a clinician’s perspective, this is useful information in balancing the diminishing returns between collecting gait cycles and inducing errors.
4. Overall, the experimental design is well-defined from a statistical perspective.

Validity of the findings

No changes from previous feedback (reproduced below)

1. The raw (open source) data and plots from the authors’ tests are provided, which agree with each other.
2. It is understandable that broad brushstrokes are difficult in this research area. Accordingly, several limitations are outlined, including potential bias due to data from consecutive running bouts, choice of ground truth value, etc.
3. Modulo the above limitations, the conclusions of this paper are solid and relevant from a theoretical and a clinical perspective.

Additional comments

None of my concerns (reproduced below) have been addressed. I cannot commend the article for publication unless they are addressed.

1. While the analysis is comprehensively presented in the figures, it is slightly difficult to compare between kinematic variables at slow and fast speeds. As this is one of the important take-aways of this paper, I would recommend showing a side-by-side comparison.
2. I would suggest that the authors investigate statistical techniques/tests to compare two distributions. I believe that adding results of a T-/Z- test between low-speed and high-speed data would improve readability and credibility.

Reviewer 3 ·

Basic reporting

The authors have improved this in their revision.

Experimental design

This aspect has been addressed adequately.

Validity of the findings

No new comments.

Additional comments

Thank you for making the recommended changes to the manuscript as appropriate.

---

## Round 0.3 · accepted · Accept

· Academic Editor

Accept

The authors have addressed all of the reviewers' comments.

Reviewer 2 ·

Basic reporting

No changes from previous feedback

Experimental design

No changes from previous feedback

Validity of the findings

No changes from previous feedback

Additional comments

- I am content with the figures added in the supplemental document, I appreciate the authors for this effort.
- I am content with the arguments they present against showing statistical tests.

I support and recommend this manuscript for publication (with the supplementary figures included).